# Nuclear Import and Export of YAP and TAZ

**DOI:** 10.3390/cancers15204956

**Published:** 2023-10-12

**Authors:** Michael Kofler, András Kapus

**Affiliations:** 1Keenan Research Centre for Biomedical Science of the St. Michael’s Hospital, Toronto, ON M5B 1W8, Canada; michael.kofler@unityhealth.to; 2Department of Surgery, University of Toronto, Toronto, ON M5T 1P5, Canada; 3Department of Biochemistry, University of Toronto, Toronto, ON M5B 1T8, Canada

**Keywords:** YAP/TAZ/Yorkie, mediated nuclear import, nuclear export, nuclear pore complex, importins and exportins, nuclear import signal and nuclear export sequence

## Abstract

**Simple Summary:**

YAP and TAZ are transcriptional activators, which play critical roles in the regulation of cell/organ growth and regeneration. Accordingly, they are important mediators and modulators of cancer formation and spread, impacting a wide range of processes from cancer stem cell differentiation through the control of proliferation, phenotype determination, and migration, till the induction of antitumor drug resistance. These widespread activities are due to their profound effect on gene expression. Importantly, to exert these physiological and pathological functions, they must enter the nucleus, while for the termination of their transcriptional effects, they must exit. However, the critical structural requirements (nuclear localization signal and nuclear export signal) as well as their passageway(s) and mechanism(s) for crossing the nuclear pore are only beginning to emerge. This review summarizes the current knowledge about the nuclear traffic of YAP and TAZ, a fundamental, novel, and potentially exploitable aspect of their biology.

**Abstract:**

Yes-associated Protein (YAP) and its paralog Transcriptional Coactivator with PDZ-binding Motif (TAZ) are major regulators of gene transcription/expression, primarily controlled by the Hippo pathway and the cytoskeleton. Integrating an array of chemical and mechanical signals, they impact growth, differentiation, and regeneration. Accordingly, they also play key roles in tumorigenesis and metastasis formation. Their activity is primarily regulated by their localization, that is, Hippo pathway- and/or cytoskeleton-controlled cytosolic or nuclear sequestration. While many details of such prevailing *retention models* have been elucidated, much less is known about their actual nuclear traffic: import and export. Although their size is not far from the cutoff for passive diffusion through the nuclear pore complex (NPC), and they do not contain any classic nuclear localization (NLS) or nuclear export signal (NES), evidence has been accumulating that their shuttling involves mediated and thus regulatable/targetable processes. The aim of this review is to summarize emerging information/concepts about their nucleocytoplasmic shuttling, encompassing the relevant structural requirements (NLS, NES), nuclear transport receptors (NTRs, karyophererins), and NPC components, along with the potential transport mechanisms and their regulation. While dissecting retention vs. transport is often challenging, the emerging picture suggests that YAP/TAZ shuttles across the NPC via multiple, non-exclusive, mediated mechanisms, constituting a novel and intriguing facet of YAP/TAZ biology.

## 1. Introduction: A Central Enigma in the Biology of YAP and TAZ

YAP and TAZ are paralogous transcriptional coactivators that are primarily regulated by the Hippo pathway [1,2,3] and the cytoskeleton [4,5,6]. They play key roles in a plethora of physiological processes—such as tissue growth, regeneration, differentiation, and matrix deposition [1,7,8,9,10,11], and, accordingly, in related pathologies, including organ fibrosis [12,13,14,15,16] and cancer [17,18,19,20].

### 1.1. Types of YAP/TAZ Contribution to Cancer

The types (and root causes) of YAP/TAZ dysregulation in cancer fall in the following three categories.

(1) Wild type YAP and/or TAZ are often activated and overexpressed in cancer cells or cancer stem cells, as observed in human malignancies, including breast, liver, colon, prostate, pancreatic, ovarian, uterine, and renal carcinomas, as well as melanoma, sarcomas, and neuronal tumors [17,18,21,22,23,24,25,26,27,28,29,30,31]. In a small subset of tumors (<10%), overexpression or activations of wild type YAP/TAZ is due to genetic alterations [18,32]. These include the amplification of YAP/TAZ genes themselves [32,33], or mutations in their upstream suppressors, primarily components of the Hippo pathway [18,32]. Accordingly, certain tumors harbor loss-of-function mutations in the core Hippo kinases (MST1/2, LATS1/2), their adaptors (SAV1), or their upstream regulators. The most prevalent of the latter are germline or somatic mutations of the Neurofibromatosis-2 (NF2) gene, encoding Merlin [34], a cytoskeletal protein and potent activator of the Hippo pathway [32]. In addition to neurofibromatosis, a hereditary tumor syndrome [34], somatic NF2 mutations are found in 30–40% of malignant mesotheliomas, and less frequently in other tumors (renal cancer and cholangiocarcinoma) [32,35,36,37,38]. Nevertheless, in the vast majority of cases, overactivation of wild type YAP/TAZ is not due to Hippo-related mutations but due to dysregulation of a large variety of signaling events affecting their transcription, stability, and transcriptional activity. Moreover, these phenomena occur not only in the tumor cells themselves, but also in the surrounding cell types, including cancer-associated fibroblasts (CAFs) and the endothelium [22,39,40,41]. These are critical for shaping the tumor microenvironment (TME) [40,42], which, in turn, promotes tumor growth and spread through mechanical (extracellular matrix) and paracrine (cytokines) cues [43,44]. In many cancers, dysregulated YAP/TAZ signaling shows strong correlation with poor survival [19,32]. Finally, it is worth noting that in addition to human malignancies, a large variety of experimental murine cancer models also revealed increased nuclear accumulation of wild type YAP/TAZ by immunostaining. These include breast cancer [45], non-small cell lung cancer [46], colorectal cancer [47], gastric cancer [48], hepatocellular carcinoma [49], pancreatic adenocarcinoma [50], cutaneous squamous cell and basal cell carcinoma [51], melanoma [52], and soft-tissue sarcoma [53].

(2) Cancer-related point mutation(s) in the coding region of YAP/TAZ proteins per se seem exceedingly rare [32]; so far, the YAP1 R331W missense mutant was reported to predispose for lung adenocarcinoma [54] and activating somatic mutations of regulatory phosphorylation sites were detected in YAP in a melanoma patient [55].

(3) The presence of various YAP or TAZ fusion proteins is the direct cause for tumor formation in a set of rare malignances, exemplified by epithelioid hemangioendothelioma [56,57,58]. The resulting oncoproteins are composed of an N-terminal portion of YAP or TAZ fused to a C-terminal sequence of another protein (e.g., TAZ-CAMTA1, YAP-TFE3); they are resistant to upstream regulatory signals and constantly nuclear, as dictated by the fusion partner.

### 1.2. Key Importance of Nucleocytoplasmic Shutting

In all the above scenarios, YAP/TAZ initiate transcriptional reprograming (YAP/TAZ signature) [29,32,59,60], primarily by activating the TEAD family of transcription factors [61,62,63,64]. This in turn impacts a large cohort of tumor-promoting processes, including stemness, epithelial-mesenchymal transition, proliferation, metabolism, invasion/metastasis, angiogenesis, and drug resistance [19,32,33,65]. In aggregate, all of the studies (Section 1.2) prompt two main conclusions, relevant to the topic of this review. The first is that in most cases, the tumor-promoting role of YAP/TAZ is associated with the dysregulation of their wild type versions; the second is that the nuclear presence of YAP/TAZ is indispensable for these effects in all tumor categories. Thus, understanding the nucleocytoplasmic traffic of YAP/TAZ, and the contributions of its various determinants, that is, cytosolic and nuclear retention and nuclear import and export, is of paramount importance.

### 1.3. Retention Models

There are two prerequisites for the nuclear accumulation of YAP/TAZ: they must be “free” (sufficiently mobile), and they must enter the nucleus (Figure 1).

The overwhelming majority of studies so far have focused on the first requirement and, accordingly, their nuclear accumulation was interpreted in the context of retention models. Indeed, the Hippo pathway is a major regulator of their cytosolic retention. When this cascade is active, MST1/2 kinase-mediated activation of LATS1/2 kinase induces phosphorylation of YAP/TAZ [1,2,3]. This results in their interaction with “cytosolic retention factors”, such as members of the 14-3-3 family, thereby preventing their nuclear accumulation [66,67]. YAP/TAZ sequestration is also regulated by large protein complexes associated with the plasma membrane and the underlying cytoskeleton. In polarized cells, these include members of the Crumbs complex, the Par complex, and tight junctions (e.g., angiomotins, Kibra, Merlin, ZO2) in the apical compartment, and the Scribble complex and adherens junctions (e.g., Scrib, α-cadherin, PTPN14, Merlin, Kibra) in the basolateral compartment. These components were shown to act as activators or scaffolds of the Hippo pathway, orchestrating the inactivation of YAP/TAZ, and/or as direct binders of YAP/TAZ (for reviews see [68,69]). The cytoskeleton was also shown to regulate the free pool, independently of the Hippo pathway: F-actin and YAP compete for the cytosolic retention factor angiomotin; thus, an increase in F-actin can liberate YAP for nuclear uptake [5,70].

Since a multitude of conditions, including the presence of extracellular ligands, the integrity of intercellular contacts, metabolic state, matrix stiffness, and other mechanical cues, as well as genetic defects can affect Hippo pathway activity and/or cytoskeletal organization [71,72,73,74,75,76], all these inputs can impact the mobile pool of YAP/TAZ. Conversely, nuclear retention is dependent on YAP/TAZ binding to intranuclear partners, such as TEAD proteins, which in turn can result in net nuclear accumulation [77,78,79,80].

Finally, it is worth noting that in addition to the classic Hippo kinase-mediated Ser phosphorylation, YAP/TAZ are subject to a variety of other posttranslational modifications [81,82,83], including Ser/Thr phosphorylation by AMPK [84], JNK [85], and MST4 [45], tyrosine phosphorylation (e.g., by Yes, Src, and Abl kinases [86,87,88,89]), acetylation [90], methylation [91], and sumoylation [92]. Many of these modifications affect localization, e.g., AMPK and the methyl transferase Set7 are negative, whereas MST4 and Abl are positive regulators of nuclear accumulation. However, in most cases (except for MST4, see Section 3.3.2) it remains to be clarified whether these modifications act on cytoplasmic/nuclear retention or alter nuclear transport as well.

### 1.4. The Fate of “Free” YAP/TAZ: Nuclear Flux

But once “free”, how do YAP and TAZ enter (and leave) the nucleus? Although their molecular mass is not far from the reported cutoff limit for passive diffusion through the nuclear pore complex (NPC), evidence is accumulating that such a mechanism (alone) is unlikely to account for their nucleocytoplasmic shuttling (see below). However, YAP/TAZ do not contain a conventional nuclear localization signal (NLS) or a prototypic nuclear export signal (NES). Thus, the possibility of mediated transport (import and export) posed intriguing questions about the identity of these sequences in YAP/TAZ. Further, recent observations suggest that in addition to the availability of YAP/TAZ for transport, mechanical forces may regulate both the shape of these molecules and their passageway, the NPC [93]. While this field is in a primordial state yet, it is becoming increasingly clear that the import of YAP/TAZ may involve multiple mechanisms. Accordingly, the purpose of this review is to summarize and organize the emerging knowledge on YAP/TAZ nuclear shuttling. First, we will provide a brief overview of the “classic” and other karyopherin-dependent nuclear transport pathways, and then discuss the different transport modalities whereby YAP/TAZ is thought to shuttle across the nuclear membrane.

## 2. Nuclear Import and Export of Proteins, a General Overview

Proteins access the nucleus through NPCs which are embedded in the nuclear envelope, a double lipid bilayer separating cytoplasm and nucleoplasm. NPCs are 60–120 MDa structures assembled from multiple copies of 30 different nucleoporins, totaling approximately 1000 proteins [94,95,96,97]. The central pore of NPCs is lined with intrinsically disordered phenylalanine-glycine (FG) repeat domains that establish a sieve-like permeability barrier with a soft exclusion size threshold [98,99,100,101,102]. In consequence, passive diffusion of proteins beyond 30–60 kD is dramatically suppressed. To overcome this size restriction and allow for nucleocytoplasmic distributions of proteins beyond simple diffusional equilibration, specialized nuclear transport receptors (NTR) exist that shuttle protein cargoes across the nuclear envelope [103,104,105,106,107,108,109]. NTR recognize nuclear import cargoes via nuclear import signals (NLS) and export cargoes through nuclear export sequences (NES) (Figure 2).

Direct interactions of the NTR with FG repeat domains allows the NTR-cargo complexes to translocate through the NPC by facilitated diffusion [102]. A sharp concentration gradient of the small Ran GTPase bound to GDP (RanGDP) vs. GTP (RanGTP) exists across the nuclear envelope, and this is the driving force for vectorial protein transport [110,111]. Mechanistically, RanGTP, which dominates in the nucleus, triggers the disassembly of incoming NTR-cargo complexes [112] (Figure 1, step 2) and the assembly of complexes designated for nuclear export [113] (Figure 1, step 4, 6). In the cytoplasm where RanGDP is prevailing, exported NTR-Cargo-RanGTP complexes dissociate, driven by the hydrolysis of bound GTP to GDP (Figure 1 step 5, 7). Low levels of RanGTP in the cytoplasm also permit the formation of NTR-cargo complexes destined for import (Figure 1 step 1). The asymmetric distribution of Ran GTPase activating proteins in the cytoplasm, boosting Ran GTPase activity, and the chromatin-bound guanine-nucleotide exchange factor RCC1, which facilitates GTP loading of Ran, maintains the steep RanGDP/RanGTP gradient. The RanGDP-specific transport receptor NTF2 replenishes Ran levels in the nucleus [114] (Figure 1, step 9, 10).

### 2.1. Nuclear Import Pathways

#### 2.1.1. Classic Protein Import

The most extensively used protein import mechanism in cells is known as the classic nuclear import pathway. It depends on importin β1, a member of the karyopherin-β type NTR family, which recognizes cargoes with classic NLS indirectly. It binds protein adaptors belonging to the importin α NTR family (Figure 2), the direct receptors for classic NLS (Table 1). In the nucleus, the importin β-importin α-cargo complexes are disassembled through RanGTP (see Section 2), and the importin α-specific (karyopherin-β type) NTR CAS/CSE1L shuttles importin α back to the cytoplasm. Classic NLS come in two major flavors: monopartite comprising one stretch of positively charged (basic) residues (Lys, Arg), or bipartite containing two stretches, separated by a linker (Table 1) [104,105,115].

Cargo specificity in the classic nuclear import pathway is attained by the seven importin α subtypes (in humans), which show differences in their cargo recognition and their expression in distinct cell types, tissue, and developmental stages. As a variation to this theme, importin β1 can bind directly to certain cargo proteins [107,115].

#### 2.1.2. Protein Import by Other Karyopherin-β Type NTR

The full complexity of nuclear import systems becomes more tangible when looking at the entire family of karyopherin-β NTR, which encompasses twenty members in humans [106] (Table 1). Besides importin β1, nine others are also involved in protein import. Of these, transportin and transportin 2 are the best characterized and recognize a non-classic, so called PY-type NLS (stretches of 20–30 amino acids harboring a C-terminal R/K/H(X)_2-5_PY sequence [105]). Another member, importin 7 (IPO7) is of particular interest since it has been recently implicated in mechanosensitive import of YAP [116]. Importin 7 can function as a genuine karyopherin-β type NTR or it acts as an adaptor protein for importin β1-mediated import. Three additional karyopherin-β members are called biportins, due to their capacity to mediate both nuclear import and export. Since a full characterization of the cargo specificities for all known NTR is still missing [106], it remains an open question whether the increasing number of “unconventional” NLS [80,117,118] are indicators for novel, undiscovered import pathways or can be assigned to one of the less well-characterized karyopherin-β NTR.

### 2.2. Nuclear Export Pathways

Four members of the karyopherin-β NTR family are involved in protein export, in addition to the bidirectional biportins. Amongst them, CRM1 appears to mediate most protein export from the nucleus [106]. CRM1 recognizes well-characterized, hydrophobic NES in cargo proteins (Table 1). However, recent experiments suggest that a large portion of CRM1-export cargoes lack a typical NES [106,119]. Conveniently, the specific CRM1 inhibitor Leptomycin B (LMB) facilitates the designation of CRM1 dependency. CAS/CSE1L is a karyopherin-β type NTR that is mainly involved in the export of importin α subtypes, as mentioned above. As for the import pathways, further research is warranted to better characterize the specificities of these export pathways. For example, it awaits elucidation whether RanBP6 and RanBP17, the last two members of the karyopherin-β NTR family, are involved at all in protein export and/or import.

### 2.3. Additional Complexities of Import and Export and the Relevance for YAP and TAZ

Several NPC components themselves have been found to directly bind cargo proteins and promote their import, independent from NTRs [120,121,122,123]. These findings blur the distinction between NPC components and NTR. Moreover, importin α subtypes were found to function in an importin β1- and Ran-independent manner in some instances [124]. Lastly, proteins can be transported through the central pore of NPCs as part of larger complexes. This process is referred to as piggyback transport and alleviates the requirement of specific localization signals for individual components of the complex [125]. All these complexities make it difficult to predict the protein transport systems responsible for cargoes that lack defined NLS and NES. Accordingly, for YAP and TAZ, notoriously lacking recognizable NLS and NES, and with molecular weights around 60 kD, the relevant nuclear import and export processes remained unknown for a long time. This has changed in recent years.

## 3. Nuclear Import of Yorkie, YAP, and TAZ

### 3.1. Import of Yorkie—A Special Case of Classic Import

As with many Hippo signaling pathway components, the first downstream effector of the pathway was discovered in Drosophila: Yorkie. Wang et al. [126] found that the N-terminal 55 amino acids of Yorkie (isoform H, which contains 23 additional amino acids at the N-terminus compared to isoform F) are essential for the protein to localize to the nucleus, when Scalloped (Sd), the Drosophila TEAD homolog was coexpressed. The N-terminus was also required to drive a Sd-reporter luciferase construct in cells and to induce an overgrowth phenotype in vivo, when expressed as a transgene in the Drosophila eye. Loss of the overgrowth phenotype upon N-terminal deletion could be rescued by fusing a classic NLS to the Yorkie construct. Using the N-terminus for pull-down experiments, they identified by mass spectrometry Drosophila importin α1 and importin β as binding partners and showed that importin α1 overexpression led to the nuclear enrichment of both the isolated N-terminus and full-length Yorkie in cells. Furthermore, they convincingly showed that importin α1 was required for the overgrowth phenotypes of the Drosophila eye in vivo, driven by Yorkie expression or hpo (MST1/2 homolog) ablation. Detailed mutational analysis revealed that Arg 15 in the N-terminus is part of the NLS in Yorkie and critical for importin α1 binding, nuclear localization, and the overgrowth phenotype. Concerning the regulation of nuclear localization, they found that the expression of hpo not only resulted in Yorkie cytoplasmic sequestration, but also reduced importin α1 binding. Similarly, expression of the constitutively GTP-loaded Ran mutant Q69L, but not the GDP-loaded T24N mutant, decreased Yorkie-importin α1 association. Therefore, this work identified the key components of the classic nuclear import pathway, importin α1, importin β, and Ran, as critical factors for Yorkie nuclear localization, albeit the discovered NLS differs significantly from the classic one. However, this import mechanism is likely not conserved in mammals, as the homologs YAP and TAZ lack this N-terminal region [127], and co-expression of KPNA6, the human homolog to drosophila importin α1, failed to affect YAP localization [126].

### 3.2. Piggyback Mechanisms

The absence of a conventional NLS in YAP/TAZ suggested that their uptake is mediated via indirect or alternative mechanisms. Of these, the most plausible is that their import still occurs in an importin α/β-dependent manner but utilizes the classic NLS of a YAP/TAZ binding partner. In other words, it is brought about by a piggyback mechanism (see Section 2.3, Figure 2C). Results of some studies compatible with such mechanisms are described below.

#### 3.2.1. MAML1/2

Members of the Mastermind-like family (MAML1-3 in mammals, homologs of Drosophila MAM) [128,129,130], were identified as key interactors and effectors of the Notch signaling pathway. Later, these nuclear proteins were found to facilitate the transcription of Wnt, Hedgehog, and NFκB pathway target genes as well [131,132,133,134], indicating that they are pivotal regulators of a wide spectrum of developmental programs. MAML proteins have a significant role in tumorigenesis, too [130,135,136]. Screening for proteins that promote the nuclear localization YAP/TAZ, Kim et al. [137] identified MAML1 and 2 whose ectopic expression induced strong nuclear accumulation of both YAP and TAZ and promoted the expression of their target genes, ANKRD1, CTGF, and CYR61. Conversely, knockdown of MAML1/2 shifted YAP/TAZ from the nucleus to cytosol. Importantly, the PPxY motif in MAML1 directly bound to the WW domain of YAP/TAZ. In addition, the deletion of the classic NLS from MAML1, as well as deletion or inactivation of the WW domain in YAP, abolished the MAML1-induced rise in nuclear YAP accumulation. MAML1 was binding equally well wild type YAP and YAP 5SA (mimicking the active, dephosphorylated form), and was also able to induce nuclear accumulation of the YAP 5SA, but not its WW mutant. This observation indicated that the effect of MAML1 on YAP is Hippo pathway-independent. Taken together, these results were consistent with a model wherein cytosolic MAML1 or 2 binds YAP or TAZ and carries the partner into the nucleus using its own classic NLS. However, the results are also consistent with MAML1/2-induced nuclear retention of YAP/TAZ imported by separate mechanisms, and this is the preferred interpretation by the authors themselves. This view is based on their observations that the kinetics of MAML1 and YAP nuclear entry are different, the former being faster than the latter. Further support for this view comes from the fact that MAML1 forms a trimeric complex with YAP/TAZ and TEAD. The authors also suppose that if import were the critical mechanism, then YAP phosphomimic mutants, which show cytosolic retention would have lower affinity for MAML1; however, this is not the case. Finally, their finding that downregulation of MAML1 does not prevent nuclear YAP accumulation in Hippo pathway-inactivated (LATS KO) cells, although it inhibits the transcriptional output of YAP, is also in line with uncoupled uptake mechanisms. Nonetheless, it cannot be excluded that a small pool of YAP/TAZ co-translocated with MAML1/2, but there is no current evidence to support this. In any case, nuclear enrichment, as well as the oncogenic properties of YAP/TAZ (proliferation, colony formation, invasion) is highly potentiated by overexpression and mitigated by the absence of MAML1. Whether this effect involves any import modulation—in addition to nuclear retention and augmented transcriptional activity—remains to be determined. The authors of this work elegantly showcase the difficulties to distinguish between retention and transport control, albeit clearly observable effects on nuclear localization and biological functions.

#### 3.2.2. Mask

Another example of possible piggyback regards the Mask family of ankyrin-repeat- and KH-domain-containing proteins, the Drosophila Mask, and it its mammalian homologs, Mask1 (ANKHD1) and Mask2 (ANKRD17) [138,139]. Mask was originally identified as a component of receptor tyrosine kinase signaling in Drosophila eye development [138], providing the initial example for the widespread roles of the family. Ankyrin repeat proteins are important scaffolds for a variety of protein–protein interactions, while KH domains exhibit RNA/ssDNA-binding [140,141,142]. Accordingly, these proteins have been implicated in multiple functions, including proliferation, differentiation, cytoskeletal organization, and transcriptional control [141,143,144,145]. Moreover, their dysregulation and role are increasingly recognized in the context of tumorigenesis/metastasis in various malignancies [139,143,144,146,147,148]. Parallel discoveries in two laboratories have revealed that Mask binds Yorkie and acts as an important cofactor strongly potentiating the transcriptional activity of Yorkie and YAP [78,149]. Subsequent studies by Sidor et al. [150] have shown that the knockdown of Mask prevents or strongly mitigates nuclear accumulation of Yorkie during the developmental stretch (flattening) of Drosophila ovarian follicle cells, and that the classic NLS in Mask is necessary for its capacity to induce the nuclear enrichment of Yorkie. The addition of the classic NLS to Yorkie bypasses the requirement of Mask expression for proper Yorkie localization. Analogous observations were made for overexpressed GFP-Yorkie in the columnar cells of the wing. Furthermore, Mask1/2 downregulation in mammalian cells (siRNA) or in organoids (from conditional Mask1/2 KO mice) reduced both the nuclear localization and the total expression of YAP. The authors interpreted these findings to mean that Mask proteins, via their classic NLS, mediate the nuclear import of Yorkie/YAP/TAZ. This is certainly plausible; however, the results are also consistent with Mask-induced nuclear retention and accumulation of Yorkie/YAP/TAZ. Indeed, overexpression of Mask1/2 results in the formation of granular aggregates of YAP both in the cytosol and in the nucleus. The authors suggest that Mask promotes clustering/polymerization and consequent colloidal phase separation, as well as the stabilization of YAP. These effects are fully compatible with (and suggestive of) a retention model. This study did not follow the kinetics of Mask vs. YAP/TAZ translocation, which could potentially differentiate piggyback transport from retention. However, Sensores-Garcia et al. [78] reported that hpo inactivation or Sd (TEAD homolog) overexpression translocated Yorkie but not Mask into the nucleus, indicating that these factors can move (accumulate) separately. Finally, the finding that an artificial classic NLS fused to Yorkie is necessary to overcome the effect of Mask depletion is curious, given that the N-terminus of Yorkie contains an NLS [126]. Arguably, retention-augmented nuclear accumulation can be mimicked by robust, energy-dependent nuclear import, driven by a strong NLS. Taken together, future studies are necessary to verify the exact nature of the effects of Mask on import, the YAP/TAZ region necessary for Mask binding, and its relationship to sequences affecting import.

#### 3.2.3. Selective Piggyback?

In principle, selective binding partners for YAP and TAZ could support selective import of these molecules. This has been proposed for the p300 acetyltransferase I (p300), which binds TAZ but not YAP [151]. Accordingly, p300 was suggested to shuttle TAZ (and its other binding partner Smad3) but not YAP into the nucleus via a piggyback mechanism, requiring the classic NLS within p300. It must be mentioned, however, that nuclear retention by the predominantly nuclear p300 could also explain these findings. In either case, p300 potentiated the fibrogenic capacity of TAZ and Smad3 in TGFβ-stimulated hepatic stellate cells which are the main mediators of liver fibrosis.

#### 3.2.4. MRTF: A Caution

It is worth mentioning that the binding of YAP/TAZ to a classic NLS-containing protein by no means guarantees piggyback-type import or even nuclear retention. In this regard, we have shown that TAZ binds to myocardin-related transcription (MRTF) partly via the interaction of the TAZ WW domain with the PPXY sequence in MRTF [13]. This was an exciting observation because it raised the possibility of uncovering a key mechanism whereby the cytoskeleton regulates the nuclear localization of YAP/TAZ. MRTF is a monomer actin-binding protein, whose classic NLS is masked by G-actin. Upon actin polymerization, G-actin dissociates from MRTF, thereby liberating its NLS and inducing nuclear accumulation [152,153,154]. Since similar cytoskeleton-remodeling stimuli promote the nuclear uptake of MRTF and YAP/TAZ, it was a plausible hypothesis that MRTF drags along YAP/TAZ to the nucleus. However, overexpression of constitutively nuclear (G-actin-binding deficient) MRTF did not redistribute TAZ to the nucleus, and MRTF downregulation did not inhibit, but facilitated nuclear TAZ accumulation. Thus, MRTF acted as a cytosolic TAZ retention factor and vice versa. These proteins antagonize each other at the level of nuclear accumulation, while once in the nucleus, they can functionally synergize, often driving the same genes via adjacent cis elements [13,155,156]. This fine-tuned regulation may establish a stimulatory threshold, above which efficient amplification occurs. From a transport standpoint, the take-home message is that co-import vs. compartmental retention (cytosolic or nuclear) is dependent on the binding partner, the stimulus, and the cellular context. For example, the association of the partners may reduce import by elevating the molecular weight or by masking the corresponding NLS (see Figure 1).

### 3.3. YAP/TAZ-Import Involving Importin α

#### 3.3.1. CSE1L/Importin α

The possibility of a direct involvement of classic import components in YAP/TAZ uptake was raised by pharmacological studies. Nagashima et al. [157] found that a TAZ inhibitor, named TI-4, potently abrogated the nuclear accumulation of TAZ without affecting its phosphorylation. Searching for the underlying mechanism of this Hippo-independent effect, they identified CSE1L (exportin-2) as the major target of TI-4. In keeping with this, CSE1L silencing also suppressed TAZ accumulation and eliminated any (additional) TI-4 effect. Both CSE1 knockdown and TI-4 reduced the rate of recovery of nuclear TAZ levels after photobleaching (FRAP), suggesting slower import. Since CSE1L is known to mediate the nuclear export of importin α5 (human KPNA1), the authors tested the potential involvement of this NTR. Both CSE1L and TAZ were found to bind importin α5, and TI-4 strengthened this interaction with the former, while it was weakened with the latter. TI-4 redistributed importin α5 from the nucleus to the cytosol, presumably by blocking its release from CSE1L. Finally, ivermectin, an inhibitor of classic (importin α/β1-mediated) import, slowed TAZ entry as measured by FRAP. CSE1L was shown to regulate YAP1 localization in a similar manner. Based on these findings, the authors proposed the following model: CSE1L exports importin α5 from the nucleus, which in turn binds TAZ in the cytosol and facilitates its import. TI-4 inhibits the dissociation of importin α5 from CSE1L, and therefore its association with TAZ; CSE1L downregulation abrogates the cytosolic availability of importin α5. This is certainly an intriguing model that warrants further investigation. Unfortunately, the impact of importin α5 downregulation was not tested in this study. A recent publication using a different cell type (RPE-1) found a mild (15%) but significant reduction in the nucleocytoplasmic ratio of YAP upon importin α5 silencing, while importin β1 silencing had no effect [116]. Thus, it remains unclear whether this pathway requires importin β1 or may represent an example for the importin α-dependent/β1-independent uptake, as mentioned in Section 2.3 (also described for casein kinase IV [124] or the Ran-independent import of importin α per se [158]).

#### 3.3.2. Importin α5/α8

Further findings supporting the role of importin α5 in YAP import, and its potential regulation, were obtained by An et al. [45]. These authors showed that MST4, a member of the mammalian sterile-like kinase family, also affects YAP localization, similarly to its classic Hippo kinase relatives MST1/2 but via a different mechanism. Namely, MST4 induces direct phosphorylation of YAP on Thr 83 (distinct form the classic Hippo cascade targets, e.g., Ser 127) and this weakens the association between YAP and importin α5 as well as importin α8 (KPNA7). The authors proposed that this results in suppressed nuclear entry and thus enhanced cytosolic accumulation. Consistent with this view, phosphomimetic (T83D) and non-phosphorylatable (T83A) YAP mutants accumulate in the cytosol and the nucleus, respectively. They also show that dephosphorylation of Thr 83 promotes YAP/TAZ target gene activation and tumorigenesis in vitro and in vivo in a gastric cancer context. Taken together, MST4-mediated Thr 83 phosphorylation of YAP negatively correlates with YAP and importin α5 association and YAP nuclear localization. A confounding factor is the location of Thr 83 within the TEAD binding domain of YAP. Phosphorylation of this site reduces TEAD binding, and it remains to be seen whether this is rather the consequence of reduced import or the cause of diminished nuclear retention.

Overall, future studies should address the exact nature, mechanism, and structural basis (YAP/TAZ sequence) of importin α5-mediated YAP import, as well as its relation to the other import mechanisms and its pathophysiological significance.

### 3.4. YAP/Yorkie Nuclear Import by a Karyopherin-β Type NTR

Very recently, a nuclear import pathway for YAP has been identified that involves importin 7 [116]. The authors of this elegant paper started off with an unbiased approach, involving isobaric labeling of cyto- and nucleoplasmic fractions together with quantitative mass spectrometry to identify proteins with mechanoresponsive (i.e., cell density-sensitive) nucleocytoplasmic shuttling. The NTR importin 7 was one of the proteins with increased nuclear presence at low confluence. Other mechanical cues such as growth on large spreading areas or on stiff substrates and mechanical stretching also increased the nuclear accumulation of importin 7, whereas pharmacological treatments that diminished actomyosin-controlled tension or mechanical uncoupling of cyto- and nucleoskeleton, via a mutation in the connecting complex (LINC complex), had the opposite effect. The researchers recognized the similarity in mechanoresponsive nuclear localization of importin 7 and YAP, and these proteins had been shown to interact in a previous publication [159]. Immunoprecipitation experiments, proximity ligation assays, interaction studies with purified proteins, and localization studies in cells confirmed that the interaction between importin 7 and YAP occurred within cells, was direct, and depended on three regions in YAP which are also critical for nuclear localization: unphosphorylated Ser 127 (the major Hippo-regulated phosphorylation site), amino acids 314–320 (resembling a proposed NLS for importin 7 [160]), and the C-terminal PDZ-binding motif (PBM), which was reported to play a role in supporting the nuclear localization of YAP and TAZ [80,161,162]. Concerning the regulation of the importin 7-YAP interaction, they found mechanical cues to operate upstream of the Hippo kinase MST1/2: high cell density or low actomyosin-controlled tension inhibited the interaction, which could be rescued by the MST1/2 inhibitor XMU-MP-1. To determine whether importin 7 was a bona fide NTR for YAP, as opposed to sequestration-based nuclear accumulation, Garcia et al. tested the capacity of importin 7 to help YAP partitioning into a hydrogel formed by recombinant FG domains and to mediate nuclear import in digitonized cells. These experiments indicated that YAP transport is Ran sensitive and depends on importin 7 and energy. Correspondingly, depletion of importin 7 using siRNAs strongly diminished nuclear accumulation of YAP/TAZ and reduced their transcriptional activity, as measured by a TEAD luciferase reporter construct and the expression of YAP/TAZ target genes ANKRD1, CTGF, and CYR61. In comparison, when individually depleting the other karyopherin-β type NTRs involved in import, including two of the three biportins, or the full set of importin α subtypes, only a slight reduction of YAP and TAZ nuclear localization was observable upon elimination of importin αs KPNA1, KPNA2, and KPNA7. This is consistent with earlier reports [45,157] implicating these importin α subtypes in YAP nuclear localization based on observed interactions (Section 3.2). Intriguingly, YAP, in return, controlled localization and function of importin 7: depletion of YAP or inhibition of the importin 7-YAP interaction by mechanical cues decreased importin 7 nuclear localization, whereas the MST1/2 inhibitor XMU-MP-1 had the opposite effect. However, unlike YAP, depletion of other importin 7 cargoes—Smad3 and Erk2—did not alter importin 7 localization. YAP also reduced the association between importin 7 and Smad3 or Erk2 and the nuclear presence of these cargoes. These intriguing findings add another mechanism to the complex functional interplay between YAP and other transcription factors such as Smad3; thus, in addition to sequestration and expression control, altered transport can also play a role. Lastly, the researchers confirmed that the interactions and functions of importin 7 were conserved in Drosophila. The importin 7 homolog Msk bound Yorkie which required unphosphorylated Ser 168 (the equivalent to YAP Ser 127) and the C-terminal residues of Yorkie. Moreover, Msk was important for the nuclear localization of Yorkie and a Yorkie-driven overgrowth phenotype in the dorsal compartment of the wing imaginal disc.

### 3.5. YAP/TAZ Nuclear Accumulation Involving Unconventional Import Routes

Our approach to investigate the process of TAZ nucleocytoplasmic shuttling [80,163] involved a molecular toolkit of fluorescent tags and sequestration systems we developed to study the mediated nuclear import and export of TAZ in cells, unbiased by contributions from passive diffusion-driven flux and retention-based compartmentalization. The tags consisted of multiple mCitrine fluorophores, as described elsewhere [100], to increase the molecular weight of TAZ beyond the exclusion size limit of the NPC and, thereby, minimize the passive passage through the central pore. Since the barrier function of the NPC displays a soft, gradual exclusion size threshold with residual entry of even large fluorophore-multimers beyond 150 kD, rather than a sharp cutoff at 60 kD [98,99,100], we utilized the mCitrine homopentamers as tag (5C) and carefully determined the ratio of fluorescence found in the nucleus and the cytoplasm (N/C) of transfected cells. With these precautions, the 5C-tag diminished passive influx sufficiently to detect even weak and highly specific import signals, however, at the cost of reduced import dynamics. These limitations were overcome by our RIS’N (Rapamycin-Induced Sequestration in the Nucleus) system. It is based on the Rapamycin-induced strong interactions between an FRB domain, fused to mCitrine and a NES (1C-NES-FRB), and a FKBP domain of a nuclear anchor construct (H2B-2xFKBP-mCherry). When the 1C-NES-FRB is fused to a fragment with NLS function, it shuttles between nucleus and cytoplasm, but the potent NES maintains a steady-state cytoplasmic localization. Rapamycin-triggered nuclear sequestration blocks export, and the dynamics of nuclear import can be unmasked and monitored by the increase in nuclear fluorescence. Equipped with these tools, and using the TAZ 4SA mutant, in which all LATS phosphorylation sites were mutated to alanine to eliminate cytoplasmic retention, we identified a 15-amino acid long region (327–341) in the TAZ C-terminus as NLS. This segment works as a bone fide import signal, in that it is both necessary and sufficient for nuclear entry. Moreover, two features of this NLS proved to be indispensable for import: a central methionine residue flanked by negative charges ([80] and submitted manuscript). These unique features distinguish the TAZ NLS (which we term M-motif because of the essential methionine) from any other identified NLS. Importantly, mutation of the critical residues abrogated import in the context of full-length (otherwise intact) TAZ as well. In addition, the three hydrophobic motifs FLxxΦ present in the NLS, region 360–365 and the C-terminal residues, cumulatively support import, but this motif is dispensable for the function of the isolated identified NLS ([80] and submitted manuscript). The mediated import of TAZ, as measured by the steady-state distribution of 5C-fusion proteins, was insensitive to Ran depletion or the expression of constitutively active Ran mutant G19V, in contrast to a 5C construct with classic NLS. Consistent with Ran independence, the identified unusual NLS does not engender a large net nuclear accumulation of TAZ over the cytosolic level in the absence of nuclear retention. The NLS is evolutionarily conserved both in YAP and TAZ. In addition, our recent studies have uncovered the presence of similar and transport-component sequences (M-motifs) in various cellular and viral proteins as well, suggesting that the TAZ NLS may be a prototypic member of a new NLS family (manuscript submitted). Taken together, our studies define an independent and functionally important import motif in YAP/TAZ, which is necessary for basal, mediated diffusion across the nuclear pore. Future studies should test its functional relationship (or the lack thereof) with other transport-affecting sequences in YAP/TAZ and the associated mechanisms.

### 3.6. YAP/TAZ Interactions with the NPC

#### 3.6.1. Nup37

A growing list of articles report interactions between YAP/TAZ and components of the NPC and show functional consequences in that the downregulation of these affect the nuclear localization and transcriptional activity of the two transcriptional coactivators [120,121,159]. One group studied the role of Nup37, a component of the NPC scaffold [164], in hepatocellular carcinomas (HCC). They found elevated expression of Nup37 in this carcinoma which promoted growth, migration, and invasion [120]. Nup37 downregulation inhibited intrahepatic metastasis in a mouse model. Given the importance of YAP-signaling in HCC, they further examined the interplay between Nup37 and YAP. The authors showed that Nup37 is critical for YAP transcriptional activity, as demonstrated by the expression of CTGF, CYR61, and Cyclin E. Nup37 expression increased nuclear YAP, and the migration-promoting effects of Nup37 expression could be suppressed by YAP silencing. Moreover, immunoprecipitations detected binding between Nup37 and YAP, which depended on the YAP WW domain. These results identified Nup37 as a significant component for YAP nuclear localization and function, promoting the authors’ conclusion that it enhances either transport or retention.

#### 3.6.2. Nup205

In a very recent publication [121], researchers concentrated on the role of YAP and TAZ in podocytes, a cell type critical for the maintenance of the glomerular filter. They identified by quantitative label-free mass spectrometry several karyopherins (IPO5, 7, 8, 9, 11, Xpo2, 5, Xpot) as well as Nup107, Nup133, and Nup205 in immunoprecipitates of both YAP and TAZ. Interestingly, Nup205, the most abundant Nup amongst the YAP/TAZ interactors, was found to be mutated in families suffering from steroid-resistant nephrotic syndrome [165]. Downregulation of Nup205 decreased YAP/TAZ nuclear localization, YAP-TEAD interaction, and the expression of YAP/TAZ target genes ANKRD1, DIAPH3, CTGF, and CYR61. Moreover, depletion of either Nup205 or YAP reduced cell proliferation and increased oxidative stress-induced cell death. On the other hand, depletion of TAZ diminished Nup205 expression and nuclear localization. Correspondingly, loss of TAZ negatively impacted YAP nuclear localization via its effects on Nup205, revealing an interesting TAZ-dependent regulation of YAP activity. The authors assigned Nup205 a direct role in YAP/TAZ shuttling, consistent with a publication that found Nup205 to enter the nucleus in a NTR- and Ran-independent manner and that identified extensive structural similarities between Nup205 and NTRs [123]. This work adds a further mechanism for YAP/TAZ import, but additional studies are warranted to better distinguish effects on transport from changes in nuclear (Nup205-mediated) retention or consequences on NPC conformations upon Nup205 depletion.

#### 3.6.3. Involvement of Other Nups?

Nup43, Nup98, Nup153, and Nup155, in addition to importin 7, were identified as YAP interactors in an article researching mechanotransduction through a Caveolin-1/YAP signaling axis [159]. The study was conducted in the context of pancreatitis, since YAP is required for pancreatitis-induced acinar-to-ductal metaplasia, and upregulation of Caveolin-1 expression decreases survival of pancreatic cancer. Nup98 and Nup155 were also amongst the top ten genes, whose siRNA mediated knockdown blunted YAP nuclear translocation. While this report did not focus on transport aspects of YAP regulation, it served as a resource for a later study [116] involving importin 7 in YAP transport (see Section 3.4).

### 3.7. Regulation of YAP/TAZ Nuclear Import

#### 3.7.1. Import Regulation by Posttranslational Modifications

Retention mechanisms are critical determinants of YAP/TAZ localization as they control the pool of “free”, transport-competent species (see Section 1.3 and Section 1.4, and Figure 1). Inherently, regulation of retention and import are tightly interlinked: phosphorylation of site Ser 127 in YAP, for example, is suggested to negatively affect importin 7 binding [116] and, at the same time, it establishes the association with 14-3-3 proteins. Similarly, phosphorylation of Thr 83 diminishes importin α5/α8 binding but might also affect interactions with TEAD [45]. However, further studies are needed to understand how the ever-growing list of YAP/TAZ posttranslational modifications (Section 1.3) affect localization and what the underlying mechanisms are.

#### 3.7.2. Import Regulation by Mechanical Forces

Transduction of signals emanating from mechanical forces emerged as an essential part of many aspects of cell (and tissue) fate and behavior in as much that it developed into the distinct field of mechanobiology [166]. In general, mechanical cues generated outside or within the cell can be sensed via force-sensitive structures in the cell periphery (cilia, focal adhesions, cell–cell contacts, individual (transmembrane) proteins) or through direct deformation of internal organelles, primarily the nucleus. The latter can result from migration through confined space or osmotic stress [167,168].

Concerning the regulation of YAP/TAZ import by mechanical forces, these cues can be transduced through biochemical signals (posttranslational modifications) which is addressed in the previous Section 3.7.1. Here, we focus on mechanical cues (stiffness, shear stress, tensile, contractile, and osmotic forces) that affect cell and nuclear shapes via transmission through the cytoskeleton-LINC complex–nuclear lamina axis or through direct nuclear deformation (in general [169,170,171,172,173,174], specific for YAP/TAZ [175,176,177]).

Novel insights into the mechanoresponsive regulation of YAP import came from the groundbreaking work of the Roca-Cusachs lab [93,109]. The group showed that mechanical cues are transmitted to the nucleus via the cytoskeleton-LINC complex axis, leading to deformation of the nuclear envelope and, thereby, widening the pore diameter of the embedded NPCs (Figure 3).

Furthermore, the mechanical forces enhanced both Ran-dependent YAP import and passive influx of control constructs. These changes in import were sensitive to the molecular weight and mechanical stability of the cargoes themselves [93,178]. The relatively small YAP (≈55 kD), being mainly composed of intrinsically disordered regions, displays low mechanical stability. Hence, YAP is perfectly fit for the mechanosensitive import. This work compellingly revealed that the cytoskeleton, in addition to its impact on the Hippo pathway activity (see Section 3.4) and direct cytoplasmic retention via angiomotins and MRTF, also modulated YAP transport itself and suggested pore dilation as the underlying mechanism. It remains to be seen how these findings can be integrated with the importin 7-mediated, mechanosensitive YAP import [116], as well as the role of identified NLS sequences.

Moreover, we could also show that mediated import of TAZ, which depended on the novel NLS we identified, was increased upon stimulation of Rho activity [80]. Unpublished work indicates that the TAZ NLS-mediated import is importin 7-independent, suggesting that nuclear deformation might stimulate import through this pathway as well.

Lastly, mathematical modeling of substrate stiffness, engagement, and dimensionality (2D vs. 3D cultures) was used to explain nuclear-cytoplasmic distribution of YAP/TAZ under different conditions [179]. This work laid out the complex regulation of YAP/TAZ localization by mechanical and geometrical properties of the cell environment as well as cell and nuclear shape. For example, the clear differences in localization in 2D vs. 3D cultures at the same substrate stiffness underlines the need to study dynamics of YAP/TAZ localization in vivo.

## 4. Nuclear Export of Yorkie, YAP, and TAZ

### 4.1. CRM1-Mediated Export

Strong support for a CRM1-mediated export pathway of YAP/TAZ stems from their observed nuclear accumulation upon treatment of cells with the CRM1-specific inhibitor LMB [80,180,181]. Using a rapamycin-induced cytoplasmic retention system, we verified the LMB-sensitive export, and identified the corresponding NES involving residues 35–38 within the TEAD binding domain of TAZ [80]. These experiments also revealed that LMB directly affected efflux dynamics, rather than affecting localization via redistribution of retention factors from the nucleus to the cytoplasm. The localization of the NES within the TEAD binding domain suggested that TEAD binding masks the NES. Moreover, binding studies revealed that 14-3-3 binding reduced TEAD association, and this could consequently unmask the NES, establishing an export-prone state of the molecule [80].

### 4.2. Piggyback Export

In addition to this direct nuclear export, a piggyback-based mechanism involving the NES of Merlin has been proposed [182]. In epithelial cells, Furukawa et al. found that the association of Merlin with the adherens junction component E-cadherin is regulated by tension in the underlying circumferential actin belt. In contrast to other parts of the cytoskeleton, this actomyosin-generated tension led to YAP suppression (reduced nuclear localization and transcriptional activity), rather than activation. Actin belt tension induced the dissociation of Merlin from the adherens junctions, increased YAP–Merlin interaction and enhanced nuclear efflux, whereas import was unaltered, as determined by LMB-time course experiments.

### 4.3. Regulation of Export

It is plausible that the nuclear export of YAP/TAZ is a highly regulated process. Using quantitative analysis of fluorescence photobleaching experiments, coupled with sophisticated mathematical modeling of transport kinetics, Ege et al. determined rates for global association and dissociation of YAP with nuclear and cytoplasmic retention factors and rates for import and export [183]. The detailed analysis identified the major regulatory pathways impinging on YAP/TAZ phosphorylation, such as the Hippo pathway and Src-family kinase activity downstream of the cytoskeleton, to regulate primarily YAP export. These are intriguing insights, adding details to the predominant import- and retention-centric views of the regulation of YAP/TAZ localization. While hugely informative, it can be added that, given the large number of known factors affecting retention and transport in a context-dependent manner, care should be taken when interpreting modeling-based analysis with its inherent simplifications.

## 5. YAP and TAZ in the Focus of Drug Therapy

### 5.1. Brief Overview of the Current State

Given their role in tumorigenesis, YAP and TAZ are attractive drug targets. However, because they mainly consist of inherently disordered regions and lack typical “druggable pockets” or enzymatic activity, their direct pharmacological inhibition is extremely challenging [184]. Therefore, apart from the use of antisense oligonucleotides directed against YAP/TAZ transcripts, either their upstream regulators or downstream effectors have been targeted. Notable examples in the first category include statins, which are widely used inhibitors of the Rho/Rho kinase pathway [73,185,186], the antidiabetic drug Metformin, which acts primarily by inducing AMPK-mediated inhibitory phosphorylation of YAP [187,188], and various tyrosine kinase (e.g., Src, Abl) inhibitors [189,190,191], which downregulate YAP/TAZ expression, stability, and nuclear localization. However, the real breakthrough has come with the development of drugs that inhibit YAP/TAZ–TEAD interaction and TEAD activity. The first compound recognized to interfere with YAP/TAZ-dependent TEAD activation was Verteporfin, a photosensitizer used in the treatment of macular degeneration [192,193]. It is, however, rather unspecific with poor pharmacokinetics and toxicity profile. Recently, several companies developed new TEAD inhibitors with high selectivity, efficacy, and improved pharmacokinetics. These drugs either target with high affinity YAP/TAZ binding sites (interface 3 and 2) in TEAD (protein–protein interaction (PPI) inhibitors), or an internal hydrophobic pocket, where TEADs are (auto)palmitoylated (allosteric palmitoylation inhibitors). Palmitoylation is necessary for TEAD activity [184,194,195]. Excellent recent reviews summarize the progress in this evolving field, describing the various drugs and their actions in different preclinical models [18,184,195,196,197,198]. For example, palmitoylation inhibitors (by Vivace Therapeutics and Dana Farber) were shown to suppress tumor growth in strictly YAP/TAZ-dependent NF2-deficient tumors, such as mesothelioma, schwannoma, and meningioma [199,200,201]. Importantly, three TEAD inhibitors, VT3989 (Vivace Therapeutics), IK-930 (Ikena Oncology), and IAG933 (Novartis) entered Phase I clinical trials [198].

### 5.2. Inhibition of Nuclear Transport in Cancer: General Mechanisms and YAP/TAZ Specific Perspectives

With the above advances in mind, the question arises whether the potential inhibition of nuclear entry of YAP/TAZ signifies a viable option. Several karyopherins, including importin β1, various importin α subtypes, and CRM1 (XPO1) are overexpressed in different malignancies [202,203,204]. Moreover, inhibitors of nuclear protein transport exert strong anti-tumor effects. For example, nuclear import (importin β1 or α/β1) inhibitors, such as ivermectin, importazole, INI-43, diaminoazole derivates, and diterpenoids show potent tumor growth-inhibitory effects in ovarian, cervical, esophageal, colorectal, and prostate cancer as well as glioblastomas, schwannomas, and hematological malignancies [204,205,206,207]. Conversely, selective inhibitors of nuclear export (SINE), e.g., selinexor that targets CRM1, are clinically used drugs in the treatment of multiple myeloma, leukemias, and sarcomas [205,207,208,209]. Perturbation of nuclear traffic interferes with tumor growth through several mechanisms, encompassing altered cycling of transcription factors, induction of cell cycle arrest, and apoptosis. Thus, in principle, targeting nuclear traffic is a viable option for cancer therapy. However, two remarks of caution must be added. First, given the large number of cargoes using these karyopherins, the selectivity of these compounds is bound to be limited, and the potential toxicity is an important caveat. In this regard, cargo-specific karyopherin inhibitors represent a major advance, and are emerging as potent antiviral agents. Second, the impact of these drugs on YAP/TAZ traffic, and the potential contribution of this to their overall anti-tumor effect remains to be elucidated. Nonetheless, the recent identification of several YAP/TAZ sequences critical for import, as well as specific karyopherins and NPC components involved in their nuclear uptake, holds realistic promise for the development of selective YAP/TAZ transport inhibitors. Indeed, some hits in high-throughput screens seem to act via such mechanisms. Thus, a better understanding of YAP/TAZ nuclear import and export may lead to the discovery of a completely new class of drugs, suppressing the action of these proteins in an orthogonal way.

## 6. Conclusions and Future Directions

Our understanding of the nuclear traffic of YAP/TAZ is in a nascent state. The emerging picture suggests that the nucleocytoplasmic shuttling of these critical regulators is brought about by multiple mediated mechanisms, which involve different sequences within the cargo itself, various NTRs, other ancillary transport partners, and distinct components of the NPC (Figure 4, Table 2). However, the sought-after synthesis is still missing. Future studies should clarify the exact relationship between proposed (and not yet identified) NLS and the relevant karyopherins, as well as the presence and significance of direct interactions of YAP/TAZ (or their novel binding partners) with the critical nucleoporins. The regulation of import and export—in terms of the availability/affinity of the NLS and NES, with respect to the posttranslational modifications of YAP/TAZ and the NTRs and via mechanical or chemical modulation of the resistance/permeability of the NPC itself, are fertile grounds for future research. Distinction between sequestration and import/export per se should also be further elucidated. However, it is important to realize that these inputs do not act in isolation; in fact, the same sequences or molecular players may affect both transport, retention, and intrinsic transcriptional activity (e.g., the NLS identified by us also contributes to transactivation, while the availability of the NES is regulated by sequestration). Finally, deciphering the transport machinery holds real promise for selectively targeting YAP/TAZ import and export. This would provide new tools to modify the activity of these important regulators and might one day contribute to the treatment of fibrosis and cancer.

## Figures and Tables

**Figure 1 cancers-15-04956-f001:**
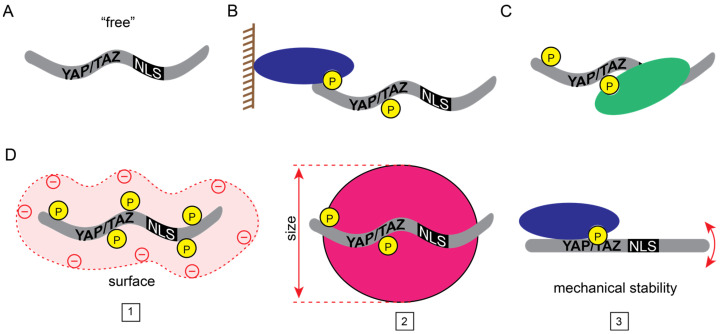
Transport states of YAP/TAZ. (**A**) “Free”, transport-competent YAP/TAZ. (**B**–**D**) Posttranslational modifications and interactions with other proteins could (**B**) tether YAP/TAZ to macromolecular complexes (e.g., plasma membrane, chromatin), (**C**) mask their transport signals (i.e., NLS or NES), or (**D**) change their physicochemical properties (1: surface charge, 2: size, 3: mechanical stability) to diminish nucleocytoplasmic shuttling.

**Figure 2 cancers-15-04956-f002:**
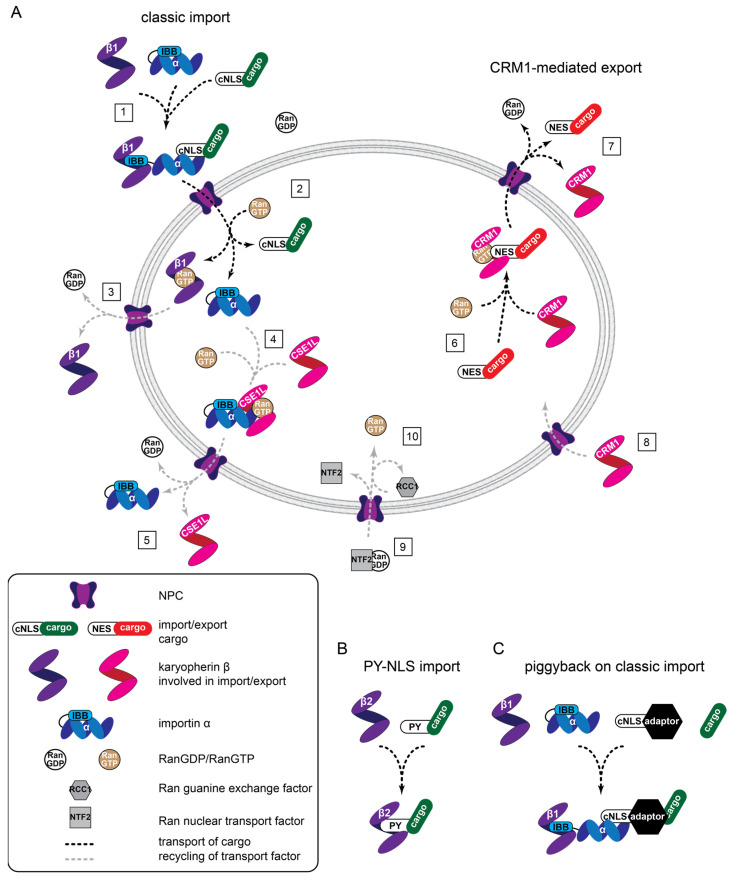
Nuclear transport machines and different transport modalities. (**A**) General overview of classic protein import (1, 2), CRM1-mediated export (6, 7), and the recycling of factors supporting transport (3–5, 8–10). (1) Assembly of a heterotrimeric import complex, NPC passage, and (2) RanGTP-driven disassembly. Importin β1-RanGTP complexes cycle back to the cytoplasm where (3) hydrolysis of GTP to GDP triggers Ran dissociation. (4) CSE1L associates with RanGTP and importin α, passes the NPC, and (5) releases importin α into the cytoplasm. (6) Similarly, CRM1 associates with RanGTP and an NES-containing cargo for (7) nuclear export, followed by (8) CRM1 reentering the nucleus. (9) RanGDP is transported into the nucleus by NTF2 and RCC1 catalysis is the exchange of GTP for GDP (10) to maintain the RanGDP/RanGTP gradient across the nuclear envelope. (**B**) Transportin-mediated import involves direct binding between karyopherin-β2 (transportin) and cargo. (**C**) Classic, piggyback import involves an NLS containing an adaptor that bridges importin α and the cargo. IBB: importin β binding.

**Figure 3 cancers-15-04956-f003:**
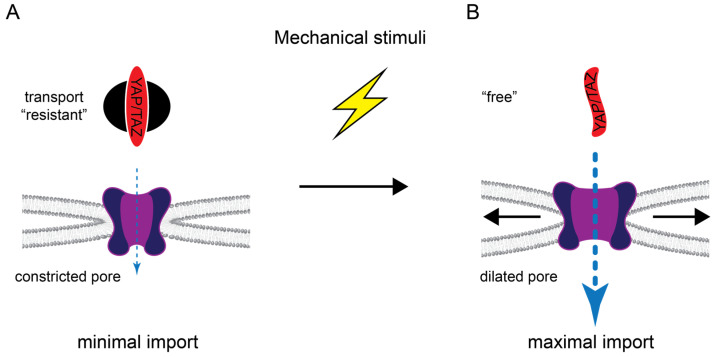
Regulation of YAP/TAZ nuclear transport by mechanical stimuli. (**A**) In the basal state, YAP/TAZ are transport “resistant” (see Figure 1) and the NPC is in a more constricted conformation, resulting in minimal/no import. (**B**) Mechanical stimuli can convert YAP/TAZ into a free state, for example, by reducing Hippo pathway activity (and thereby reducing YAP/TAZ phosphorylation) and/or by inducing cytoskeletal rearrangements that reduce cytoplasmic retention via angiomotins and MRTF. Mechanical forces are also transmitted to the nuclear envelope to increase tension which provokes dilation of the NPC and nuclear import.

**Figure 4 cancers-15-04956-f004:**
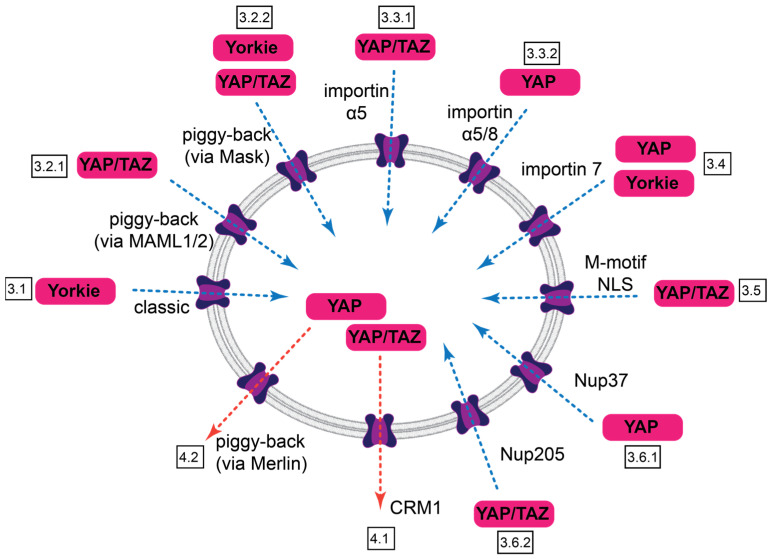
Summary of nuclear import and export pathways for Yorkie, YAP, and TAZ. Import routes are indicated by blue dotted arrows, export routes by red ones. Identified features of the transport and boxed section numbers are indicated.

**Table 1 cancers-15-04956-t001:** List of karyopherin-β NTR (modified from [106]).

Karyopherin-β	Other Names	Function	Cargo Recognition/NLS
Importin-β1	KPNB1	Import	Monopartite cNLS ^1^ K-K/R-X-K/R Bipartite cNLS ^1^ K/R-K/R-X_10–12_-K/R_3/5_
Importin-β2	Transportin, KPNB2	Import	PY-NLS
Importin-β2b	Transportin 2, KPNB2B	Import	PY-NLS
Importin 12	Transportin 3	Import	RS-NLS Phospho-RS or RD/RE dipeptide repeats
Importin 4	RanBP4	Import	Folded domains
Importin 5	KPNB3	Import	IK-NLS K-V/I-X-K-X_1–2_-K/H/R Folded domains
Importin 7	RanBP7, IPO7	Import	Folded domains
Importin 8	RanBP8	Import	Folded domains
Importin 9	RanBP9	Import	Folded domains
Importin 11	RanBP11	Import	Folded domains
Importin 13	RanBP13	Biport	Folded domains
Exportin 4	XPO4	Biport	Folded domains
Exportin 7	XPO7	Biport	Folded domains
CRM1, Exportin 1	XPO1	Export	CRM1 NES Φ-X_2–3_-Φ-X_2–3_-Φ-X-Φ Φ-X-Φ-X_2_-Φ-X-Φ Φ-X_2_-Φ-X_3_-Φ-X_2_-Φ Φ-X_3_-Φ-X_2_-Φ-X_3_-Φ Φ-X-Φ-X_2–3_-Φ-X_2–3_-Φ
CAS/CSE1L	XPO2	Export	Importin α
Exportin-t ^2^	XPO3	Export	RNA structures
Exportin 5	XPO5	Export	RNA; phospho-NES
Exportin 6	XPO6	Export	Folded domains
RanBP6	NA	NA	NA
RanBP17	NA	NA	NA

^1^ cNLS are recognized via importin αs; ^2^ not involved in protein transport; Φ: hydrophobic amino acid; NA: not applicable, unknown.

**Table 2 cancers-15-04956-t002:** Transport characteristics for Yorkie, YAP, and TAZ.

Cargo	Cargo Recognition	Binding Protein	Mechanism of Action	Reference
Yorkie	Unusual classic NLS in N-term.; involving R15	Importin α1, importin β	Classic import	[126]
YAP/TAZ	WW domain	MAML1/2	Piggyback import or nuclear retention	[137]
Yorkie/YAP/TAZ	?	Mask1/2	Piggyback import	[150]
YAP/TAZ	?	Importin α5	?	[157]
YAP	?	Importin α5 and α8	Classic import	[45]
Yorkie/YAP	S127, 314–320, PBM	Importin 7	Karyopherin-β-mediated import	[116]
YAP/TAZ	327–341	?	Novel, M-motif-dependent import	[80] and submitted
YAP	WW domain	Nup37	Undefined import or nuclear retention	[120]
YAP/TAZ	?	Nup205	Nup205-mediated import	[121]
YAP/TAZ	35–38	CRM1	CRM1-mediated export	[80]
YAP	?	Merlin	Piggyback export	[182]

?: unknown.

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
