# Peer review of "Nuclear Import and Export of YAP and TAZ"

_cancers, 2023, doi:10.3390/cancers15204956_

Round 1

Reviewer 1 Report

Although there are many reviews on the function of YAP/TAZ, there are not many reviews focusing on the molecular mechanism of its nuclear transport. Therefore, the authors' review summarizing the data to date on this subject is of great interest. This paper is worthy of publication in its entirety,  but there are some points that could be improved upon. Please refer to the comments below.

Minor comments

1. In section 1.1, the authors list three categories of the types of YAP/TAZ dysregulation in cancer. Somatic loss or inactivation of NF2 is frequently seen in spontaneous schwannomas and mesotheliomas, but is YAP/TAZ dysregulation due to alterations in the Hippo pathway gene not a fourth category?

2. Figure 2 would be kind to the reader if a brief explanation could be added to the legend.

3. Is there a reason why IPO7 is not labeled as importin 7 like other importin family proteins? It might be good to add a column of Gene name in Table 1.

Author Response

We thank the reviewers for their useful and constructive comments that greatly helped us to improve the manuscript. We supplemented the Review with a substantial amount of additional relevant information, addressing essentially all the queries, and included two new figures. 

1. Excellent point. We included a section about the genetic alterations of the Hippo pathway or its upstream regulators, including the role of NF2 in various tumors. Since these changes affect the activation/nuclear accumulation of wild type YAP/TAZ, we elaborated on these mechanisms in the first category.   

2. We included new figures, so the old Fig 2 is now Fig 4. Importantly, this figure now indicates the various sections that contain the detailed description of each represented mechanism.

3. We unified the labeling as suggested.

Thank you again. 

Reviewer 2 Report

The review entitled ' Nuclear import and export of YAP and TAZ’ is very interesting and gives an overview of the Hippo signalling members YAP and TAZ. Overall, the article describes the current understanding of the pathway and signalling mechanisms to the nucleus.

The authors could improve the review article.

1.     The authors summarized the upstream targets of YAP and TAZ on page 2; an illustrative picture would be more appealing to the readers in the review.

2.     The authors can describe about the post translational mechanisms of YAP and TAZ, such as phosphorylation or ubiquitination and other mechanisms.

3.     The main transport mechanism of YAP and TAZ to the nucleus is through other proteins, the authors could shed some light on possible intervention mechanisms to inhibit this nuclear translocation.

4.     As overexpression of YAP and/or TAZ is commonly observed in various cancers, The article needs to include a separate section describing the possible mechanism to target these proteins using inhibitors or other therapeutics.

5.     The authors could describe about the cross talk with other signalling pathways, It would be interesting to know if any adaptor proteins are involved in this cascade that promotes the nuclear export except importins and other transporter proteins described in the article.

6.     The authors can include a section on emerging targets of the pathway.

7.     The authors need to include if any clinical trials are ongoing on these targets.

8.     The discussion is too short, can the authors discuss more broadly in relation to cancer.

On page 2, line 50. the authors need to check the sentence ''YAP/TAZ overexpression is due to gene duplication/amplification''.

Author Response

We thank the reviewers for their useful and constructive comments that greatly helped us to improve the manuscript. We supplemented the Review with a substantial amount of additional relevant information, addressing essentially all the queries, and included two new figures. 

1. The Hippo pathway and its upstream regulators are subject to many excellent recent reviews (many of them cited), summarizing and illustrating these general aspects. Since our focus is bound to be selective – concentrating on nuclear import and export aspects of these phenomena -, we thought that another figure about this well-represented facet here might not be necessary or conducive. We also added two new figures to the manuscript, and we wished to avoid information overload and to minimize not directly relevant information.

2. This has been added as proposed to section 1.3 and their relevance in nuclear accumulation is now emphasized.

3. Since these points (3, 4, 6, 7) address various aspects of the same theme, we answer them together. A whole section on therapy has been added to the manuscript (Section 5). This includes a) previous and current approaches in targeting YAP/TAZ action via influencing signaling or YAP/TAZ-TEAD interaction/action; b) nuclear import and export in general with regard to oncotherapy; and c) the potential of targeting YAP/TAZ nuclear transport, as a future direction.  Clinical trials are also mentioned.

5. The role of critical adaptors in transport (e.g. Merlin as a Hippo activator and YAP/TAZ export regulator), the major transport-affecting posttranslational modification of YAP/YAZ by the Hippo pathway and beyond, as well as the details of mechanosignaling have been added to the revised version.

8. We addressed this request by including new sections and substantial detailed discussions throughout.

Language remark:

The sentence has been revised and clarified.

Reviewer 3 Report

In this review, Kofler and Kapus discuss the possible mechanisms of nuclear import and export of YAP/TAZ. Though this nucleoplasmic shuttling of TAZ and YAP is known for some time now, comprehensive discussion on how these paralogues proteins move in or outside the nucleus still remains a topic of debate. Here the authors provide a vivid explanation of the possible roles of classical and unconventional methods of nuclear import proteins, piggyback methods of export and import and other factors regulating this process.

In general, this review is comprehensive and well documents the emerging literature in a relatively new area. As such, I have a few suggestions to make it more exciting for the field.

Specific comments

1) The concept of mechanical stretching versus pressure application based nuclear deformation and YAP/TAZ localization needs more extensive discussion, Probably, a figure illustrating the plausible mechanisms would make it even better (mentioned in brief in section 3.7 and also in 3.4).

2) What about common and unconventional inhibitors of nuclear import-export? A table summarizing their impact on YAP/TAZ will also improve this version. For instance they mention about MST1 inhibitor on IPO7-YAP interactions and likewise the inclusion of more Hippo pathway inhibitors in this discussion would add more value.

3) A section on how nuclear-cytoplasmic shuttling has been reported in various mouse models of cancer that utilized either transgenic YAP or TAZ expression and or their deletion in different organs would additionally be more informative.

Author Response

We thank the reviewers for their useful and constructive comments that greatly helped us to improve the manuscript. We supplemented the Review with a substantial amount of additional relevant information, addressing essentially all the queries, and included two new figures. 

1. We have included two figures to detail the impact of mechanical inputs on the nuclear pore and the mechanical stability of the cargo (Fig 1 and 3). We substantially extended the section on mechanical signaling, alluding to the role of the LINC complex and the cytoskeleton as direct or indirect regulators of the NPC.

2. We included a whole new section (Section 5) on pharmacological targeting of YAP/TAZ as well as nuclear import and export in general with regard to cancer therapy (please see our answers to Reviewer 2, too).  This addition definitively helped us to highlight the relevance of the topic.

3. We added a section (within 1.3) about the numerous animal models, in which enhanced YAP/TAZ activation and nuclear accumulation was documented in a cancer context. We also pointed out that further animal studies are needed to investigate the dynamics and mechanisms of YAP/TAZ nuclear accumulation in vivo.

Round 2

Reviewer 2 Report

The review article 'Nuclear import and export of YAP and TAZ' is very interesting and authors greatly described the overall pathways of YAP and TAZ signaling and the current developments in the field.